# Safety and pharmacokinetics-pharmacodynamics of a shorter tuberculosis treatment with high-dose pyrazinamide and rifampicin: a study protocol of a phase II clinical trial (HighShort-RP)

David Ekqvist,[1] Anna Bornefall,[2] Daniel Augustinsson,[3] Martina Sönnerbrandt,[2] Michaela Jonsson Nordvall,[4] Mats Fredrikson,[5] Björn Carlsson,[6] Mårten Sandstedt,[7] Ulrika S H Simonsson,[8] Jan-Willem C Alffenaar,[9,10,11] Jakob Paues,[12] Katarina Niward [12]

For numbered affiliations see end of article.

**Correspondence to**
Dr Katarina Niward;
katarina.niward@liu.se

## ABSTRACT

**Introduction** Increased dosing of rifampicin and pyrazinamide seems a viable strategy to shorten treatment and prevent relapse of drug-susceptible tuberculosis (TB), but safety and efficacy remains to be confirmed. This clinical trial aims to explore safety and pharmacokinetics-pharmacodynamics of a high-dose pyrazinamide-rifampicin regimen.

**Methods and analysis** Adult patients with pulmonary TB admitted to six hospitals in Sweden and subjected to receive first-line treatment are included. Patients are randomised (1:3) to either 6-month standardised TB treatment or a 4-month regimen based on high-dose pyrazinamide (40 mg/kg) and rifampicin (35 mg/kg) along with standard doses of isoniazid and ethambutol. Plasma samples for measurement of drug exposure determined by liquid chromatography tandem-mass spectrometry are obtained at 0, 1, 2, 4, 6, 8, 12 and 24 hours, at day 1 and 14. Maximal drug concentration ($C_{max}$) and area under the concentration-time curve ($AUC_{0-24h}$) are estimated by non-compartmental analysis. Conditions for early model-informed precision dosing of high-dose pyrazinamide-rifampicin are pharmacometrically explored. Adverse drug effects are monitored throughout the study and graded according to Common Terminology Criteria for Adverse Events V.5.0. Early bactericidal activity is assessed by time to positivity in BACTEC MGIT 960 of induced sputum collected at day 0, 5, 8, 15 and week 8. Minimum inhibitory concentrations of first-line drugs are determined using broth microdilution. Disease severity is assessed with X-ray grading and a validated clinical scoring tool (TBscore II). Clinical outcome is registered according to WHO definitions (2020) in addition to occurrence of relapse after end of treatment. Primary endpoint is pyrazinamide $AUC_{0-24h}$ and main secondary endpoint is safety.

**Ethics and dissemination** The study is approved by the Swedish Ethical Review Authority and the Swedish Medical Products Agency. Informed written consent is collected before study enrolment. The study results will be submitted to a peer-reviewed journal.

## Strengths and limitations of this study

► This phase II clinical trial of patients with drug-susceptible pulmonary tuberculosis (TB) will provide data on safety and early bactericidal activity in relation to drug exposure of an experimental shorter TB regimen based on high-dose pyrazinamide and rifampicin.

► The patients' drug exposure will be related to individual *Mycobacterium tuberculosis* minimal inhibitory concentrations, exploring pharmacokinetics-pharmacodynamics indices during high-dose pyrazinamide and rifampicin.

► Opportunities for early model-informed precision dosing after the first dose will be explored by pharmacokinetic modelling.

► A limitation of the study is the small sample size, due to an extensive and resource demanding study protocol, although this is in accordance with phase II studies and will partly be compensated for by using rich data.

**Trial registration number** NCT04694586.

## INTRODUCTION

Current treatment of drug-susceptible tuberculosis (TB) is based on recommendations by the WHO and originates from the 1970s.[1 2] During the first 2 months (intensive phase) daily doses of isoniazid 5 mg/kg, rifampicin 10 mg/kg, pyrazinamide 25 mg/kg, and ethambutol 15 mg/kg are given, followed by a continuation phase of isoniazid and rifampicin.[1] This 6-month multidrug regimen is intended to prevent acquired drug resistance, therapy failure and relapse.[3] Although

effective, the long treatment duration increases the risk of non-compliance. Hence shorter and more effective regimens with preserved safety profile are needed as part of the global strategy to eliminate TB. Several attempts to shorten TB treatment below 6 months have been unsuccessful due to increased frequency of relapse.[4–7] However, a pooled analysis of patient-level data from 4-month fluoroquinolone-containing regimens identified non-inferiority compared with standard treatment in participants with minimal non-cavitary disease.[8] Recently, a 4-month rifapentine-based regimen containing moxifloxacin for treatment of pulmonary TB was non-inferior to standard 6-month TB regimen in terms of TB disease-free survival 12 months after randomisation.[9]

Accumulating data from clinical trials exploring higher doses of rifampicin[10–13] have shown increased bactericidal activity with potential to shorten TB treatment corroborated by results from pharmacokinetic-pharmacodynamic (PK-PD) modelling and simulation.[14 15] Lately, pyrazinamide has attained increasing attention due to its potent sterilising activity with the ability to eliminate intracellular resting and slowly replicating *Mycobacterium tuberculosis* (*Mtb*) so-called 'persisters'.[16] Once the drug was added to standard TB treatment in the early 1980s the treatment length could be shortened from 9 to 6 months without increase in relapse.[2 17] Efforts to further shorten TB treatment from 6 to 4 months by simply prolonging the duration of pyrazinamide was unsuccessful because of increased frequency of relapse.[18] Observational studies of more than 100 patients with pulmonary TB[19 20] and a recent meta-analysis[21] have shown that unsuccessful treatment outcome was associated with suboptimal pyrazinamide exposure. PK studies in patients with pulmonary TB have shown lower than recommended drug concentration levels[22] with standard dosing of both rifampicin (42%–80%) and pyrazinamide (27%–53%).[23 24] In a large PK-PD study evaluating patient-level data from three phase II clinical trials with pulmonary TB patients where the majority had cavitary disease, participants received rifampicin (range 10–35 mg/kg), pyrazinamide (range 20–30 mg/kg) and two companion drugs.[25] The results showed that pyrazinamide's induction of culture negativity increased with augmented drug exposure. Hepatotoxicity was rare and not associated with pyrazinamide exposure. Furthermore, PK modelling suggested that a massive dose of pyrazinamide (>4500 mg) or increasing both pyrazinamide and rifampicin doses would be required to reach microbiological targets associated with treatment-shortening such as 90%–95% culture conversion rate by 2 months of treatment.[25] Rifampicin doses of at least three times the standard dose combined with isoniazid, pyrazinamide and ethambutol was safely administered to 63 patients with pulmonary TB for 12 weeks.[11] Rifampicin 40 mg/kg was recently identified as the highest tolerated dose with dose limiting factors reported as gastrointestinal disorders, pruritus, hyperbilirubinaemia and jaundice.[26] However, the safety of simultaneously administered high-dose pyrazinamide and rifampicin remains

to be explored. Our study aims to investigate PK-PD data, safety and tolerability of a 4-month first-line treatment with combined high dosing of pyrazinamide and rifampicin to a selected group of patients with non-advanced pulmonary TB.

## METHODS AND ANALYSIS
### Study design
We are conducting a multicentre open-label randomised controlled phase II study. The study protocol conforms with Standard Protocol Items: Recommendations for Interventional Trials, an international format for interventional trials.[27 28]

### Study setting
The study is carried out at infectious disease departments in six hospitals in Sweden, three tertiary level hospitals and three secondary level hospitals. The number of hospital beds in the hospitals range from 310 to 1340. The incidence of TB in Sweden is 4.75/100 000 (2019) and the vast majority (85%) of patients are migrants that have acquired their infection prior to arriving to Sweden.[29] The number of new culture positive pulmonary TB cases during 2019 was 93 in total within the regions served by the six hospitals included in the study.[29] A majority of the study participants will be admitted to a hospital ward as per standard of care for isolation of patients with smear positive pulmonary TB. Participants not considered in need of hospital admission or isolation will be followed by frequent outpatient visits as well as hospital admission on day 1 and 14 to facilitate the frequent PK-sampling.

### Study participants
A total number of 40 evaluable patients will be included. Study inclusion and exclusion criteria are listed in table 1.

### Study outline
The overall study outline is shown in figure 1.

After providing informed consent patients are screened for eligibility to participate in the study based on the inclusion and exclusion criteria (table 1). A review and registration of the patient's clinical status and symptoms is made, and any pre-existing symptoms and findings are recorded for future reference as potential adverse event (AE) from high-dose pyrazinamide and rifampicin can overlap with the symptoms of active TB. Provided all the inclusion criteria and none of the exclusion criteria are fulfilled according to the responsible physician, the study participant is electronically randomised in a 1:3 ratio to either open-label standard first-line TB treatment for 6 months or to a shortened treatment of 4 months in the intervention group. The higher doses utilised in the intervention group are 40 mg/kg of pyrazinamide (for 8 weeks) and 35 mg/kg of rifampicin (for 4 months) and with standard dosing of isoniazid (5 mg/kg for 4 months) and ethambutol (15–20 mg/kg for 8 weeks) (table 2).

**Table 1** Inclusion and exclusion criteria in HighShort-RP

| Inclusion criteria | Exclusion criteria |
|---|---|
| Adult (≥18 years) | Treated for TB within the last year. |
| Pulmonary TB confirmed by PCR and/or culture for *Mtb* | Molecular or phenotypic resistance to any of the first-line oral drugs |
| Eligible for first-line TB treatment, but not yet started treatment | Miliary TB or advanced pulmonary TB with sputum smear grade three or advanced TB on chest X-ray according to National Tuberculosis Association (1961)[30] defined by one or more of the following radiological findings:<br>► Cavitation of >4 cm diameter.<br>► Disseminated lesions of slight to moderate density which extend throughout the total volume of more than one lung (uni- or bilateral).<br>► Radiological signs of dense and/or confluent lesions affecting >1/3 of one lung or in both lungs combined |
| Hiv-negative | Any other infectious disease of such significance that urgent treatment is required |
| BMI >17 | Ongoing treatment with other medication where treatment with rifampicin can interact with the medication and compensatory drug adjustments cannot be made |
| Written informed consent to participate in the study | Previously known allergy to any of the first-line drugs or other contraindications according to the Summary of Product Characteristics of rifampicin, isoniazid, pyrazinamide and ethambutol |
| Fertile women must agree to use a barrier contraceptive during the study and have a negative pregnancy test on inclusion | Immunocompromised defined as primary or secondary immunodeficiency including immunosuppressive treatment such as chemotherapy, immunomodulating agents, corticosteroids equivalent to ≥15 mg prednisolone |
|  | Known liver disease or aspartate aminotransferase or alanine aminotransferase >1.5 times the upper normal level |
|  | Heart failure (NYHA class III and IV) |
|  | Kidney failure (eGFR <50 mL/min) |
|  | Dysregulated diabetes mellitus |
|  | Alcohol or drug abuse |
|  | Pregnancy or breast feeding |
|  | Bodyweight <35 or >90 kg |

BMI, body mass index; eGFR, estimated glomerular filtration rate; *Mtb*, *Mycobacterium tuberculosis*; NYHA, New York Heart Association; TB, tuberculosis.

Dose adjustment based on changes in body weight will be evaluated after 2 weeks of therapy.

Clinical evaluation of the participants and disease severity is assessed and documented according to TBscore II (table 3) as well as with chest X-ray.[30 31]

First-line drug concentration measurements in plasma are determined at 0, 1, 2, 4, 6, 8, 12 and 24 hours after drug intake on day 1 and day 14. Maximal drug concentration ($C_{max}$) and area under the concentration-time curve ($AUC_{0-24h}$) are assessed by non-compartmental analysis using the linear trapezoidal method.[32] The potential for early model-informed precision dosing of high-dose pyrazinamide and rifampicin will be explored by applying PK modelling on drug concentrations obtained after the first dose vs after 14 days of treatment. Induced sputum samples are collected on day 0, 5, 8, 15 and week 8.

## Collection of safety measures and follow-up
All study participants are monitored throughout the study as per routine care. Additional study-specific visits and monitoring by ECG, blood sampling, sputum collection and radiology is outlined in figure 1. Participants in the study will be provided with a diary to record drug intake and to note any suspected side effects from the treatment. Safety data are collected during the intensive phase by weekly AE questionnaire interviews and from the patient diary, the medical record and by frequent laboratory monitoring for blood chemistry tests as displayed in figure 1. During the continuation phase the same safety routine is applied every second week. AEs are graded based on Common Terminology Criteria for Adverse Events V.5.0. A serious AE (SAE) needs to be reported within 24 hours after the study team has been notified and is defined as death of any cause, life-threatening AE and AE that results in prolonged or new hospital admission or permanent disability. For safety reasons, AE considered to be related to TB treatment and causes cessation of pyrazinamide and/or rifampicin, needs to be reported in the same manner as SAE. Based on these reports a decision is made if the SAE requires any further action in terms of trial continuation. Any new concomitant medication is recorded and evaluated to check for potential interactions with the TB treatment. As an additional safety measure, sputum samples collected during the intensive phase will be used to evaluate the effect of combined high-dose treatment on early bactericidal activity (EBA) by analysis of time to positivity (TTP) automatically detected in BACTEC MGIT 960 (MGIT) for mycobacterial liquid culture. Adequate microbiological response is defined as sputum culture conversion between day 0 and 15 or difference in TTP between day 0 and 5 by at least 25 hours, day 0 and 8 by 50 hours or day 0 and 15 by 75 hours based on unpublished data of TTP-dynamics observed in our observational cohort study[24] and the literature.[33–35]

Study participants randomised to the intervention group will be followed for a total of 28 months to record any signs of relapse. Study participants randomised to standard of care will have a follow-up visit 6 to 12 months after completion of treatment and any signs of relapse beyond that point will be retrieved by data collection from the Swedish national TB registry 24 months after treatment completion. In addition to registration of relapse, final treatment outcome is recorded according to WHO definitions.[36]

## Bioanalytical methods
### Drug concentration measurement
Blood is collected in 6 mL sodium-heparin blood tubes, centrifuged at 2000 g for 10 min within 1 hour from sampling. Aliquots of plasma is frozen in −70°C pending analysis, which will be made every 3 months[37 38] at the

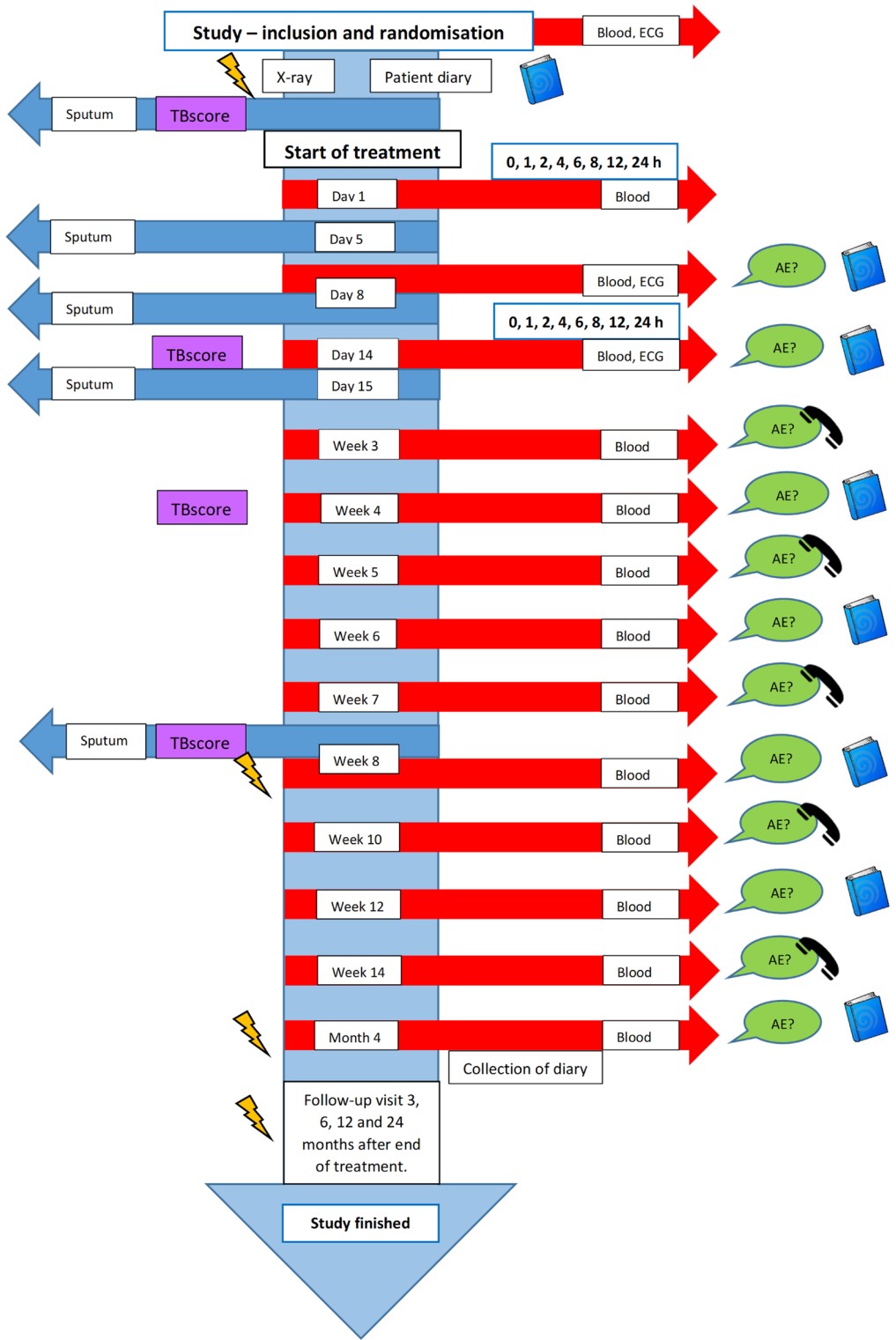

**Figure 1** The study outline in the flow chart arrow reflects study activity identical for the intervention and control group. After 4 months treatment, the intervention group is followed up according to the follow-up information found in the bottom of the flow chart arrow, meanwhile the control group is followed up according to routine care (not included in the picture). Study participants are provided with a drug diary to record concomitant drugs, food intake and potential side effects during the treatment. Sputum samples are collected regularly during the study to assess time to positivity in BACTEC MGIT 960 for mycobacterial liquid culture. Rich blood sampling for measurement of drug concentrations is collected on the day of the first dose and after 14 days of treatment. Changes in clinical status is assessed by a clinical scoring tool (TBscore II) in addition to evaluation by chest X-ray and/or thoracic CT. Safety data are monitored closely during the first four study months and is based on urgently reported serious adverse events (AEs), AE questionnaire, medical records, regularly surveillance of blood chemistry and during the first 2 weeks analysis of ECG. The final treatment outcome, according to WHO definitions and occurrence of relapse, is registered for both study arms after treatment completion. TB, tuberculosis.

**Table 2** Treatment in control and intervention group

| Drug | Control group | Intervention group |
|---|---|---|
| Pyrazinamide | 20–30 mg/kg/day for 8 weeks | 40 mg/kg/day for 8 weeks |
| Rifampicin | 10 mg/kg/day for 6 months | 35 mg/kg/day for 4 months |
| Isoniazid | 5 mg/kg/day for 6 months | 5 mg/kg/day for 4 months |
| Ethambutol | 15–20 mg/kg/day for 8 weeks | 15–20 mg/kg/day for 8 weeks |

Approved fixed dose combination tablets will be used during the treatment period and in the intervention group combined with single tablets of pyrazinamide and rifampicin to reach target doses of pyrazinamide 40 mg/kg (range 36–43 mg/kg) and rifampicin 35 mg/kg (33–37 mg/kg) according to weight-based tables included in the study protocol.

Department of Clinical Pharmacology, Linköping University Hospital, Sweden (accredited ISO/IEC 17025:2018). Total drug concentration of isoniazid, acetyl isoniazid, rifampicin, pyrazinamide, and ethambutol will be determined by preparing samples with organic solvent protein precipitation in accordance with Fang et al and Han et al.[39 40] The samples will be analysed with liquid chromatography on a reversed phase column ACQUITY UPLC BEH C18 Column, 130 Å, 1.7 μm, 2.1 mm X 50 mm a BEH C18 VanGuard pre-column and column in-line Filter (Waters Corporation, Milford, USA) and with tandem-mass spectrometry MS detection using a tandem quadrupole mass spectrometer (Xevo TQ MS, Waters) in positive electrospray ionisation mode with the following quantification traces; isoniazid 137.96>121.04, acetyl isoniazid 180.0>121.0, rifampicin 823.30>791.416, pyrazinamide 123.95>54.1, and ethambutol 205.1>116.15. The analytic range for isoniazid is 0.25–8 mg/L, acetyl isoniazid 0.25–8 mg/L, rifampicin 2.5–80 mg/L, pyrazinamide 2.5–80 mg/L, and ethambutol 0.25–8 mg/L. The drug concentration analysis method is validated according to EMA guideline on bioanalytical method validation.[41] For the quality controls at 0.75 mg/L and 6 mg/L for isoniazid, acetyl isoniazid and ethambutol the inter-day accuracy is between 95.8% and 99.9% and with a precision between 7.7% and 10.5% and for pyrazinamide and rifampicin quality controls at 7.5 and 60 mg/L

the accuracy is between 92.3% and 97.6% and precision between 9.3% and 11.8%. Intra-day at 0.75 mg/L and 6 mg/L for isoniazid, acetyl isoniazid and ethambutol gave accuracy between 86.7% and 95.6% and a precision of 0.6%–5.2% and for pyrazinamide and rifampicin at 7.5 and 60 mg/L an accuracy between 92.3% and 108.8% and precision of 2.2%–7.5%.

## Microbiology
### Time to culture positivity

Sputum samples are treated according to routine and are carried out at a biosafety level 3 laboratory at the accredited TB laboratory of the Department of Clinical Microbiology at Linköping University Hospital, Sweden. In short, following treatment of sputum with sputolysine, samples are centrifuged and resuspended in 1.5% NaOH-SDS with 2 mm glass beads and vigorously stirred using a vortex mixer. After neutralisation by 0.09% sulfuric acid samples are washed in phosphate buffer and added to MGIT-tubes according to the instructions by the manufacturer. The MGIT-tubes are incubated using BACTEC MGIT 960 (Becton Dickinson, Lakes, NJ, USA) at 37°C for up to 42 days. When growth is detected TTP is automatically recorded, *Mtb* isolates are stored at −70°C awaiting minimal inhibitory concentration (MIC) determination.

### Minimum inhibitory concentrations

Previously stored *Mtb* isolates (−70°C) from study participant obtained before treatment initiation will be used for MIC determination of first-line drugs. MIC determination will be performed for rifampicin, isoniazid and ethambutol by broth microdilution in accordance with the EUCAST reference method for *Mtb*, and when applicable additional group A drugs as displayed in figure 2.[42] MIC is defined as the lowest antibiotic concentration without growth when the 1:100 diluted controls show visual growth. *Mtb* H37Rv ATCC 27294 will be included as an internal quality control in each batch. Breakpoints established by the Clinical and Laboratory Standards Institute are used for drug susceptibility testing (DST) of isolates for rifampicin, isoniazid, and ethambutol in broth microdilution plates (1, 0.12 and 8 mg/L, respectively).[43] Routine DST of first-line drugs and MIC determination of pyrazinamide are performed in MGIT as previously described with WHO-based critical concentrations of antibiotics for routine DST (rifampicin 0.5, isoniazid 0.1, pyrazinamide 100, and ethambutol 5 mg/L) and by using

**Table 3** Parameters used for the TBscore II[31]

| Parameters | Points assigned |
|---|---|
| Self-reported | |
| Cough | 1 |
| Dyspnoea | 1 |
| Chest pain | 1 |
| Anaemia (pale conjunctiva) | 1 |
| BMI (body mass index) <18 | 1 |
| BMI <16 | 1 |
| MUAC (middle upper arm circumference) <220 mm | 1 |
| MUAC <200 mm | 1 |

TB, tuberculosis.

**Figure 2** Plate outline for MIC determination by broth microdilution using the EUCAST reference method. dH$_2$O, distilled water; EMB, ethambutol; EUCAST, European Committee on Antimicrobial Susceptibility Testing; GC, growth control; INH, isoniazid; LEV, levofloxacin; LIN, linezolid; MIC, minimal inhibitory concentrations; PZA, pyrazinamide; RIF, rifampicin.

serial twofold dilutions of antibiotics for pyrazinamide MIC testing.[44] Pyrazinamide will also be included in the reference method (pH 5.9) as an internal validation (figure 2).

**Data analysis plan**

Study monitoring is performed regularly by the regional support unit for clinical trials, Forum Östergötland, at all study sites before study opening, during recruitment period and at study closure. Obtained study data are entered in electronic case record forms and managed by using REDCap (research electronic data capture)[45 46] hosted at Linköping University and at time of data analysis transferred to a database for statistical processing.

The primary outcome is AUC$_{0-24h}$ of pyrazinamide in the intervention arm and the main secondary outcome is the incidence and severity of AEs. The distribution of AUC$_{0-24h}$, C$_{max}$ and PK-PD indices (AUC$_{0-24h}$/MIC, C$_{max}$/MIC) will be visualised separately for the two study arms. Difference in drug exposure will be explored by Mann-Whitney's U test. A p≤0.05 will be considered statistically significant. The PK-PD indices based on drug exposure at day 14 for the investigational products (pyrazinamide and rifampicin) will be correlated to tentative PK-PD targets derived from the literature on human studies such as the suggested AUC$_{0-24h}$ >363 mg*hour/L and AUC/MIC >8.42 for pyrazinamide.[47] Associations between drug exposure and liver laboratory assessments will be analysed by using Spearman rank correlation. The incidence of minimum grade two AE and SAE will be presented in frequency tables and correlation graphs on exposure-tolerability. Nonlinear mixed effects modelling will be utilised to derive a population PK model for high-dose pyrazinamide. A model-informed precision dosing approach will be derived in order to be able to predict optimised dose given information from early PK sampling. An earlier

developed model-informed precision dosing approach for high-dose rifampicin[48 49] will be applied to evaluate the accuracy in the high-dose predictions. The sample size of 10 participants in the control group and 30 in the intervention group will enable sufficient descriptive PK data of the combined high-dose pyrazinamide and rifampicin regimen accounting for the expected larger PK-variability in this group compared with the control group.[49] Applying EBA as one of the safety endpoints, we estimate 30% difference in EBA between the study arms (80% power, p=0.05), based on previous unpublished data of TTP-dynamics observed in our observational cohort study[24] combined with EBA data from the literature.[10 11 50] Randomisation is performed in blocks stratified by gender and will continue until 10 individuals in the control arm and 30 individuals in the experimental arm have completed PK sampling day 14. The randomisation scheme was done with the statistical programme R V.4 (the R foundation) by an independent statistician, uploaded into the CRF-programme REDCap[45 46] and blinded to the person who includes patients. Resistance to first-line drugs is rare in Sweden and multidrug resistant (MDR) TB accounted for 1.8% of all new TB cases 2019[29] and the majority of the drug resistant cases are detected early. Thus, late exclusions from the study based on detected drug resistance, are expected to be negligible. A modified intention-to-treat analysis will be performed for patients who may miss follow-up in addition to a per-protocol analysis.

**Patient and public involvement**

Patients were not involved in the development of the research plan nor the implementation of this clinical trial. We aim to this study results at a conference where patient representatives or patient communities are present.

Furthermore, patients will be involved in a future phase III trial based on the results of this study.

## DISCUSSION

There is strong urge for a shortened treatment of drug-susceptible TB. In this phase II clinical trial (High-Short-RP) of a shorter experimental TB regimen based on high-dose pyrazinamide and rifampicin, tolerability and safety will be investigated along with in-depth PK-PD analysis including determination of individual MICs. In a PK-PD modelling study published just after our study opened for recruitment, the relationship between pyrazinamide exposures and efficacy or hepatotoxicity was assessed on pooled retrospective data from three clinical interventional trials conducted by the Tuberculosis Trials Consortium and Pan African Consortium for the Evaluation of Antituberculosis Antibiotics Networks.[25] Simulations identified two potentially treatment-shortening strategies restricted to currently used first-line drugs whereof one was to optimise drug exposure of rifampicin and pyrazinamide in parallel. The authors stated that whether or not an enhanced multidrug regimen based on high-dose rifampicin (eg, ≥35 mg/kg) and higher-dose pyrazinamide (eg, 30–40 mg/kg) will be sufficient to significantly reduce the length of TB treatment must be explored prospectively, with attention to safety and tolerability. Results from our clinical trial will be indicative whether the combined high-dose rifampicin (35 mg/kg) with pyrazinamide (40 mg/kg) along with standard doses of isoniazid and ethambutol, is tolerable for patients. Furthermore, the PK-PD data retrieved from the study can be included in predictions of effective treatment-shortening regimens launched in larger RCTs evaluating high-dose sterilising regimens with non-inferiority criteria compared with standard of care. Recently, a large RCT of pulmonary TB patients showed that the efficacy of a 4-month rifapentine-based regimen containing moxifloxacin was non-inferior to standard 6-month TB regimen in terms of TB disease-free survival 12 months after randomisation.[9] Our long-term goal is to achieve maximum output of the first-line drugs and if possible, to avoid a fluoroquinolone-based treatment-shortening regimen.

The sample size of this phase II study is small as the study protocol is extensive and resource demanding. Each participant is closely monitored for safety in addition to frequent PK sampling during the first 14 days of the study allowing a detailed description of PK parameters. The clinical scoring system TBscore II used in the study to grade disease severity and assess treatment response was originally developed and validated for use in low-income settings and therefore scoring is based on clinical signs rather than laboratory results.[31]

Personalised medicine based on disease severity and dose-adjustments guided by therapeutic drug monitoring to achieve individual maximum exposures of TB drugs is promising, in particularly as the variation of drug exposures is extensive during high-dose rifampicin treatment.[10 15] This highlights the value of measuring drug exposure and the prerequisites for early model-informed precision dosing based on PK data obtained after the first dose will be explored in this phase II clinical trial.

## ETHICS AND DISSEMINATION

This study has been approved by the Swedish Ethical Review Authority (approval number 2020-00761) and the Swedish Medical Products Agency (approval number 5.1-2019-105207 and EudraCT number 2019-003721-25). Prior to the study opening, a designated study team of nurses, physiotherapists (training of study nurses in collection of induced sputum), laboratory staff and clinical doctors participated in training sessions of the study protocol which included study-specific procedures and ethical considerations. Study initiation meeting and the main training sessions were led by the coordinating principal investigator assisted by the support unit for clinical trials Forum Östergötland. Physicians and nurses in the study team have documented training in Good Clinical Practice (GCP) and the study is performed in accordance with GCP and the Declaration of Helsinki. Informed consent is obtained from each study participant by a physician prior to any study-specific activity. Inclusion of patients in the study is restricted to the available set of written informed consent translated into selected languages. In addition and when required, an interpreter will be used for providing study information and to translate during study visits. Study participants are free to withdraw from the study at any time and will at that point be offered treatment according to standard of care. The safety of combined high-dose pyrazinamide (40 mg/kg) and rifampicin (35 mg/kg) has not been studied in humans before. However, TB treatment with rifampicin 35 mg/kg given to 63 patients for 12 weeks has been proven safe.[11] Furthermore, data were recently presented on exposure-tolerability of 2 weeks treatment with rifampicin 40 mg/kg (n=15) and 50 mg/kg (n=17) where the dose of 40 mg/kg was safe and tolerable.[26] In early studies on lengthy treatments of pyrazinamide beyond 8 weeks, increased liver markers, jaundice and arthralgia were associated with higher doses of pyrazinamide.[51] In a meta-analysis comparing rates of AEs between different doses of pyrazinamide, the frequencies of hepatotoxicity were 0.042 (95% CI 0.026 to 0.067) for 30 mg/kg, 0.055 (95% CI 0.031 to 0.094) for 40 mg/kg and 0.098 (95% CI 0.047 to 0.193) for 60 mg/kg, suggesting a considerable proportion of hepatoxicity rates may be idiosyncratic rather than dose-dependent.[52] According to a meta-analysis (n=4490), liver toxicity did not seem to be dose-dependent until pyrazinamide exposures exceeded a weekly AUC of 5000 mg*hour/L[53] compared with an AUC of 363 mg*hour/L on standard dosing.[47] Recently, a patient-level data analysis where patients received rifampicin (range 10–35 mg/kg), pyrazinamide (range 20–30 mg/kg), plus two companion drugs, found liver toxicity to be rare (3.9% with grade 3 or higher liver

function tests, LFT), and no relationship was observed between pyrazinamide $C_{max}$ and LFT levels.[25] Furthermore, currently used doses of 30–40 mg/kg pyrazinamide daily as part of MDR-TB treatment are well tolerated.[54] In summary, we consider it safe for patients to participate in our clinical trial and study participants will be closely monitored regarding potential side effects including weekly measurement of LFT during the first 8 weeks when high-dose pyrazinamide and rifampicin are combined. As this is a phase II clinical trial exploring increased dosing of already approved and widely used drugs, the review authorities did not require a data safety and monitoring board.

Regardless of study results, we plan to present our data in international conferences and to publish our results in a peer-reviewed journal. Any significant protocol amendments will be reported to the ethical review authority and medical products agency.

**Author affiliations**
$^1$Department of Infectious Diseases, Region Östergötland, Linköping University, Linköping, Sweden
$^2$Department of Infectious Diseases, Region Östergötland, Linköping, Sweden
$^3$Department of Infectious Diseases, Region Östergötland, Norrköping, Sweden
$^4$Department of Clinical Microbiology, and Department of Biomedical and Clinical Sciences, Linköping University, Linköping, Sweden
$^5$Forum Östergötland, Linköping University, Linköping, Sweden
$^6$Department of Clinical Pharmacology, and Department of Biomedical and Clinical Sciences, Linköping University, Linköping, Sweden
$^7$Department of Radiology in Linköping, and Department of Health, Medicine and Caring Sciences, Linköping University, Linköping, Sweden
$^8$Department of Pharmaceutical Biosciences, Uppsala University, Uppsala, Sweden
$^9$School of Pharmacy, The University of Sydney Faculty of Medicine and Health, Sydney, New South Wales, Australia
$^{10}$Sydney Institute for Infectious Diseases, The University of Sydney, Sydney, New South Wales, Australia
$^{11}$Westmead Hospital, Sydney, New South Wales, Australia
$^{12}$Department of Infectious Diseases, and Department of Biomedical and Clinical Sciences, Linköping University, Linkoping, Sweden

**Acknowledgements** We thank all members of the study team at the Infectious Diseases Clinic of Östergötland, former research nurse Lisa Gunnarsson at the Department of Infectious Diseases at Linköping University Hospital, initiated local principal investigators and the research coordinator Lina Malm at Forum Östergötland, Linköping University.

**Contributors** The study design was developed within the TB research group at Linköping University Hospital led by JP and KN with support from J-WCA and USHS on the pharmacokinetic part of the study. Statistician MF assisted in statistical selection of study dimensions and developed the block randomisation procedure. AB, JP and KN wrote the study protocol and the applications to ethical review authority and medical product agency. KN is the coordinating principal investigator of the clinical trial and research nurse in lead is MSö. KN developed case record forms (CRFs), standard operating procedures (SOPs), led study initiation meeting with training sessions together with DE and assisted by DA, MSö, MJN, MSa and JP. BC is the contact person at the laboratory of the Department of Clinical Pharmacology and MJN at the TB laboratory of the Department of Clinical Microbiology. DE and KN wrote the first draft of the manuscript. All authors revised and approved the final version of the manuscript.

**Funding** This work was supported by the Swedish National Research Council (KN, grant number 2019-05912), the Swedish Heart Lung Foundation (JP, grant number 20200302), the Research Council of south-eastern Sweden (JP, grant number FORSS-931028, KN, FORSS-964535) and the Region Östergötland ALF grant (JP, grant number RÖ-888331).

**Disclaimer** The funders had no role in any of the stages of the study.

**Competing interests** None declared.

**Patient consent for publication** Not applicable.

**Provenance and peer review** Not commissioned; externally peer reviewed.

**ORCID iD**
Katarina Niward http://orcid.org/0000-0002-5290-5165

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
