## [Reviewer comments · BMJ Open]

ARTICLE DETAILS

TITLE (PROVISIONAL)	Safety and pharmacokinetics-pharmacodynamics of a shorter tuberculosis treatment with high-dose pyrazinamide and rifampicin – a study protocol of a phase II clinical trial (HighShort-RP)
AUTHORS	Ekqvist, David; Bornefall, Anna; Augustinsson, Daniel; Sönerbrandt, Martina; Nordvall, Michaela Jonsson; Fredrikson, Mats; Carlsson, Björn; Sandstedt, Mårten; Simonsson, Ulrika; Alffenaar, Jan-Willem; Paues, Jakob; Niward, Katarina

VERSION 1 – REVIEW

REVIEWER	Rolla, Valeria Cavalcanti Fundacao Oswaldo Cruz
REVIEW RETURNED	06-Sep-2021

GENERAL COMMENTS	This is a pharmacokinetics pharmacodynamic study to evaluate high doses of rifampicin isoniazid for a four months treatment of tuberculosis in a selected population. The study is interesting and based in literature findings Some points need to be clarified 1) Abstract -The authors state that shortening of TB treatment has failed, however, a published study of Dorman's and coworkers was recently published and showed that rifapentine associated with moxi, isoniazid and pyrazinamide for 4 months showed good results so is better to use other word than failed -The dose optimization that authors are referring is an increase in the regular dose, I think would be better to say this clearly -Please spell tuberculosis before use TB -This clinical trial aims to explore safety and pharmacokinetics-pharmacodynamics of a high-dose pyrazinamide- rifampicin regimen to shorten TB treatment. I understood that efficacy will be one of the aims, and if I'm correct should be included. If you are planning to check the outcomes, the cure rates and relapse is a measure of efficiency -"Clinical outcome is registered according to WHO definitions in addition to occurrence of relapse after end of treatment". WHO has recently updated the definition so is better to mention the year of the document or even say the outcomes of interest to make the abstract as clear as possible ARTICLE SUMMARY -Ethics and dissemination – "Informed written consent is collected before Participation", I would say before any study procedure or enrollment as participation is vague Introduction
---

	At the beginning the authors used TB and after tuberculosis was used. Better to decide what is the term you prefer to use and use throughout the protocol You have to mention the Dorman's study (https://www.nejm.org/doi/full/10.1056/NEJMoa2033400) that showed to be non-inferior to RHZE, I know this is a protocol already approved and is hard yo change, but I would recommend to do it as soon as you make an amendment Table 1. Inclusion and exclusion criteria in HighShort-RP. -How would you deal with the high-risk population? On a paper included in your references (Imperial et al Nature medicine) discussed the fact some patients are less proud to be treated with the short course regimen you are proposing "we have shown that a smear grade of 3+ (you have excluded) and the presence of cavitation on chest radiographs at baseline define a hard-to treat phenotype, constituting 34% of the study population (1,162 of 3,405 participants), and this group may require longer durations of treatment than the current standard 6-month regimen to achieve the highest cure rates feasible". You have excluded sputum samples with 3+ but is not clear what was the criterium for the ones with cavitation on X-rays. You have said "according to National Tuberculosis Association, 1961" but I don't know what this document says. Moreover, is a very old document, so would rather say what this document states for a better understanding of non-national readers Ethics and dissemination -What is the role of physiotherapists in your study? -I haven't seen any mention to a data safety and monitoring board (DSMB), are you planning to have one following the study since the beginning? If not, please write a justification Table 2. Treatment in control and intervention group. I'm curious to know how the study team will compose the treatment in terms of tablets for the intervention group. Treatment in the study you are proposing is based on weight. Do you have tablets formulation to compose the correct dose according to weight? For a patient with 60kg for example, using 40 mg of pyrazinamide, the dose would be 2,400 mg, and for rifampicin 60kg 35mg/kg/day would be 2,100. Could you describe the tablets you have to compose the daily dose? And include all doses according to weight of participants in the intervention? Table 3. Parameters used for the TBscore II (30). Why you decided to use a clinical parameter for anemia? A simple hemogram with hemoglobin and hematocrit would be more precise, what the study team thinks? Data analysis plan I'm a bit concerned with the number of participants in the study, due to the exclusion criteria defined by the study group. Have you made an estimate based on local recruitment and the criteria you have defined? -I'm a bit confused by these two summaries, abstract and article summary...Is this part of BMJ Open policy?
--	--

REVIEWER	Laohapojanart, Nisa Prince of Songkla University
REVIEW RETURNED	12-Dec-2021

GENERAL COMMENTS	all of my comment were in the for "peer review only" file (left side) additional of answer NO 3. Author might add modified intention to treat analysis for patients who missed follow up. 8. reference 54, 55 and 56 were not relate to the message 11. line 367 -375 were not related to this manuscript. 12. Please give reason of small sample size 4. For patient with SAEs the author did not give details to manage with.
---

VERSION 1 – AUTHOR RESPONSE

Reviewer: 1

Dr. Valeria Cavalcanti Rolla, Fundacao Oswaldo Cruz

Comments to the Author:

This is a pharmacokinetics pharmacodynamic study to evaluate high doses of rifampicin isoniazid for a four months treatment of tuberculosis in a selected population. The study is interesting and based in literature findings

Some points need to be clarified

1) Abstract

-The authors state that shortening of TB treatment has failed, however, a published study of Dorman's and coworkers was recently published and showed that rifapentine associated with moxi, isoniazid and pyrazinamide for 4 months showed good results so is better to use other word than failed

Response: We have omitted the first line of the abstract to avoid stating that the shortening of TB treatment has failed in view of the recently published study by Dorman and co-workers.

-The dose optimization that authors are referring is an increase in the regular dose, I think would be better to say this clearly

Response: We agree and have changed accordingly (Line 34).

-Please spell tuberculosis before use TB

Response: We have corrected this in the abstract when the abbreviation is used the first time.

-This clinical trial aims to explore safety and pharmacokinetics-pharmacodynamics of a high-dose pyrazinamide- rifampicin regimen to shorten TB treatment. I understood that efficacy will be one of the aims, and if I'm correct should be included. If you are planning to check the outcomes, the cure rates and relapse is a measure of efficiency

Response: We are grateful to the reviewer pointing out this important issue. To avoid ambiguities, we have rephrased the aim in the abstract (Line 36-38) and highlighted the primary and secondary

outcomes of the study in the abstract (Line 53-54) and in the method section this message has been moved to a more appropriate subsection (Line 279-280). Efficacy in terms of cure rates and relapse is not included in the main primary or secondary endpoints of this study. This phase II clinical trial is too small to have the power to evaluate any difference in cure rates and relapse since standard treatment of drug-susceptible TB has very high disease-free survival. The results from this phase II study will be indicative if this high dose regimen is a promising way forward to shorten the treatment and include the regimen in a large RCT, where efficacy in terms of disease-free survival is the primary endpoint.

-“Clinical outcome is registered according to WHO definitions in addition to occurrence of relapse after end of treatment”. WHO has recently updated the definition so is better to mention the year of the document or even say the outcomes of interest to make the abstract as clear as possible

Response: We thank the reviewer for this recommendation and have included the year of the updated version (World Health Organization. Definitions and reporting framework for tuberculosis 2013 revision, updated December 2014 and January 2020, Geneva: World Health Organization) in the running text in the abstract (Line 52) since references is not allowed.

ARTICLE SUMMARY

-Ethics and dissemination – “Informed written consent is collected before Participation”, I would say before any study procedure or enrollment as participation is vague

Response: We agree this can be expressed more clearly. The text has been changed accordingly (Line 56-57).

Introduction

At the beginning the authors used TB and after tuberculosis was used. Better to decide what is the term you prefer to use and use throughout the protocol

Response: TB is now used throughout the manuscript (except for full names such as National Tuberculosis Association) but the first time the abbreviation is used, tuberculosis has been spelled out.

You have to mention the Dorman’s study

(<https://eur01.safelinks.protection.outlook.com/?url=https%3A%2F%2Fwww.nejm.org%2Fdoi%2Ffull%2F10.1056%2FNEJMoa2033400&data=04%7C01%7Ckatarina.niward%40liu.se%7C%7C%7C637755996932624480f97208d9c3b341ac%7C913f18ec7f264c5fa816784fe9a58edd%7C0%7C0%7C637755996932626154%7CUnknown%7CTWFpbGZsb3d8eyJWIjoiMC4wLjAwMDAiLCJQIjoiV2luMzliLCJBTiI6Ikl1haWwiLCJXVCi6Mn0%3D%7C3000&sdata=ROkhdPKuQD26rjfZZXgqpniBgOc25uShwyhi3bHKg2s%3D&reserved=0>) that showed to be non-inferior to RHZE, I know this is a protocol already approved and is hard yo change, but I would recommend to do it as soon as you make an amendment

Response: Indeed, the study by Dorman et al is highly relevant and needs to be mentioned which is now done both in the introduction (line 88-91) as well as in the discussion (line 374-376).

Table 1. Inclusion and exclusion criteria in HighShort-RP.

-How would you deal with the high-risk population? On a paper included in your references (Imperial et al Nature medicine) discussed the fact some patients are less proud to be treated with the short course regimen you are proposing “we have shown that a smear grade of 3+ (you have excluded) and the presence of cavitation on chest radiographs at baseline define a hard-to treat phenotype, constituting 34% of the study population (1,162 of 3,405 participants), and this group may require longer durations of treatment than the current standard 6-month regimen to achieve the highest cure rates feasible”.

You have excluded sputum samples with 3+ but is not clear what was the criterium for the ones with cavitation on X-rays. You have said "according to National Tuberculosis Association, 1961" but I don't know what this document says. Moreover, is a very old document, so would rather say what this document states for a better understanding of non-national readers

Response: This is indeed an important issue. We have added a more detailed explanation of the radiological criteria to help the reader to understand the criteria used (Table 1). We agree that these criteria are now very old but to our knowledge, it is still used in clinical studies although AI-based algorithm are undergoing large-scale evaluation.

Ethics and dissemination

-What is the role of physiotherapists in your study?

Response: Physiotherapists have educated and trained the study nurses in the procedure of induced sputum and may also occasionally assist the study nurses with collection of induced sputum samples from study participants. This has been further clarified in line 403.

-I haven't seen any mention to a data safety and monitoring board (DSMB), are you planning to have one following the study since the beginning? If not, please write a justification

Response: No data safety and monitoring board (DSMB) is used in the study as it is not required for this phase II clinical trial exploring increased dosing of already used drugs and this is approved by the Swedish Ethical Review Authority and the Swedish Medical Products Agency. We have clarified this issue in line 435-437. As the study focuses on safety, a rigorous protocol regarding detection and management of adverse events (AE/SAE) is set up with frequent blood samples in addition to clinical assessments and questionnaires to detect side-effects in the study population. Study investigators are obligated to inform the sponsor of the study of any SAE within 24 hours (Line 196-200). In our Swedish study protocol, we have explained that there will be no safety interim analysis, but in case of 3 study participants experiencing ≥ 3 grade 3 AE or 1 study participants a grade 4-5 AE all possible or likely related to the high-dose regimen, the sponsor will summon a predefined expert panel (not involved in the management of study participants) to discuss the continuing of the trial (information added in line 200-202). In addition, an annual safety report will be sent to the Swedish medical products agency listing any SAE and SUSAR during the time period.

Table 2. Treatment in control and intervention group.

I'm curious to know how the study team will compose the treatment in terms of tablets for the intervention group. Treatment in the study you are proposing is based on weight. Do you have tablets formulation to compose the correct dose according to weight? For a patient with 60kg for example, using 40 mg of pyrazinamide, the dose would be 2,400 mg, and for rifampicin 60kg 35mg/kg/day would be 2,100. Could you describe the tablets you have to compose the daily dose? And include all doses according to weight of participants in the intervention?

Response: The treatment will be given using fixed dose combination pills composed of rifampicin 150mg/isoniazid 75mg/pyrazinamide 400mg/ethambutol 275mg, which will be given according to weight during the first two months to both the control and intervention group. For the intervention group an addition of tablets containing pyrazinamide (250-500mg) and rifampicin (150-300mg) will be used to reach the target doses. During the continuation phase a fixed dose combination pill composed of rifampicin 150mg/isoniazid 75mg will be used, and for the intervention group addition of tablets containing rifampicin (150-300mg) will be used. Dosing will be weight-based as stated in table 2 and correlated to the closest possible weight-based dosing using the available tablets. Dosing per body weight in the intervention arm will be 33-37 mg/kg for rifampicin and 35-43 mg/kg for pyrazinamide. We have added a short description of this in the table legend (Line 173-176). The fixed dose

combination pills used in the study are approved by the Swedish Medical Product Agency, thus the bioavailability data is reviewed, and the batches of pills are checked according to standard rules.

We unfortunately don't have the needed space in the article to state all doses according to weight. However, they are available in the complete Swedish study protocol.

Table 3. Parameters used for the TBscore II (30).

Why you decided to use a clinical parameter for anemia? A simple hemogram with hemoglobin and hematocrit would be more precise, what the study team thinks?

Response: The TBscore II is clinically validated for use in low-income settings and therefore don't include any laboratory parameters. Although measuring hemogram in our setting would be more feasible, that would need a new validation of the TBscore. We have included this motivation in the discussion section (Line 385-388). We are familiar with using this score from our former study by Niward et al published in JAC 2018 (doi:10.1093/jac/dky268).

Data analysis plan

I'm a bit concerned with the number of participants in the study, due to the exclusion criteria defined by the study group. Have you made an estimate based on local recruitment and the criteria you have defined?

Response: This is indeed an important question, which is a concern that is shared by the study team. Although we applied a multicenter design, the setting where the study is performed is a low prevalence TB area. An estimation has been made based on historical number of patients with TB in the uptake area of the study population including all six study sites (Linköping, Norrköping, Jönköping, Kalmar and Karolinska Stockholm with its two sites Solna and Huddinge) as described in line 133-134, thus 93 cases of culture positive pulmonary TB cases were registered during 2019. Approximately one third of these cases may be eligible for the study. The TB cohort in Sweden tend to present with less advanced disease compared with high-endemic areas, as noticed in our observational study including the same study sites (Niward et al, JAC 2018), which is also the target group for this study. Efforts have also been made with translation of study information to several languages to be able to include patients who are not native Swedish (or English) speakers to optimise the possibility of inclusion. Inclusion rate must, in our opinion, be balanced with the safety of the participants which in part explains the sample number and detailed exclusion criteria.

-I'm a bit confused by these two summaries, abstract and article summary...Is this part of BMJ Open policy?

Response: The headings and summaries are part of BMJ Open design criteria for submission of protocol papers and we wanted to stick to the recommendations of the journal.

Reviewer: 2

Dr. Nisa Laohapojanart, Prince of Songkla University

Comments to the Author:

all of my comment were in the for "peer review only" file (left side)

additional of answer NO

3. Author might add modified intention to treat analysis for patients who missed follow up.

Response: We thank the reviewer for the suggestion of adding modified intention-to-treat analysis. In the Swedish study protocol, we have a more detailed description on this matter. We will perform a per-protocol analysis and in addition a modified intention-to-treat analysis where we have the

possibility of using data from patients who by any reason is excluded or missed follow-up. We have added this information in the manuscript (Line 308-309).

8. reference 54, 55 and 56 were not relate to the message

Response: We have tried to rephrase the section where references 54-56 are used to make them fit better in the message. We have now kept the Dorman reference (previous ref 54, now ref 9) and removed the others which were more related to PK/PD.

11. line 367 -375 were not related to this manuscript.

Response: In this version, we have introduced the findings of Dorman already in the introduction as it was suggested by another reviewer which may provide a better link to the proposed study protocol. We have also clarified lines 367-375 in an effort to clarify the relations to the present manuscript (Line 374-378).

12. Please give reason of small sample size

Response: The study aims to give a detailed description of PK/PD data using combined high dose rifampicin and pyrazinamide treatment for the first time in humans. Each participant is closely monitored during the study and frequent blood samples for measurement of drug concentration are taken during the first 14 days. This makes the study labour-intense and this level of detail of each participant would be hard to achieve for a much larger number of participants and usually not needed for the purpose of a phase II clinical trial. A section to elaborate on this question has been added to the manuscript (line 383-385) in addition to the existing lines 294-300.

4. For patient with SAEs the author did not give details to manage with.

Response: Any potential adverse event will be assessed regarding intensity and grading by the local investigator and treated according to clinical practice. Reporting of SAE is described in line 196-200. Study investigators are obligated to inform the sponsor of the study of any SAE within 24 hours who will then decide if the SAE warrants further action in terms of trial continuation. This information has been added to the manuscript (line 200-202). In addition, an annual safety report will be sent to the Swedish medical products agency listing any SAE and SUSAR during the time period.

Reviewer: 1

Competing interests of Reviewer: I have no competing interests

Reviewer: 2

Competing interests of Reviewer: no

VERSION 2 – REVIEW

REVIEWER	Rolla, Valeria Cavalcanti Fundacao Oswaldo Cruz
REVIEW RETURNED	28-Jan-2022
GENERAL COMMENTS	thanks for your clarification. The paper looks much better now and I don't have any other question to raise